# Differential but Concerted Expression of HSD17B2, HSD17B3, SHBG and SRD5A1 Testosterone Tetrad Modulate Therapy Response and Susceptibility to Disease Relapse in Patients with Prostate Cancer

**DOI:** 10.3390/cancers13143478

**Published:** 2021-07-12

**Authors:** Oluwaseun Adebayo Bamodu, Kai-Yi Tzou, Chia-Da Lin, Su-Wei Hu, Yuan-Hung Wang, Wen-Ling Wu, Kuan-Chou Chen, Chia-Chang Wu

**Affiliations:** 1Department of Urology, Taipei Medical University-Shuang Ho Hospital, New Taipei City 23561, Taiwan; 11579@s.tmu.edu.tw (K.-Y.T.); 20500@s.tmu.edu.tw (C.-D.L.); 10352@s.tmu.edu.tw (S.-W.H.); 15334@s.tmu.edu.tw (W.-L.W.); kuanchou@s.tmu.edu.tw (K.-C.C.); 2Department of Medical Research and Education, Taipei Medical University-Shuang Ho Hospital, New Taipei City 23561, Taiwan; d508091002@tmu.edu.tw; 3Department of Hematology and Oncology, Cancer Center, Taipei Medical University-Shuang Ho Hospital, New Taipei City 23561, Taiwan; 4TMU Research Center of Urology and Kidney, Taipei Medical University, Taipei City 11031, Taiwan; 5Graduate Institute of Clinical Medicine, College of Medicine, Taipei Medical University, Taipei City 11031, Taiwan; 6Department of Urology, School of Medicine, College of Medicine, Taipei Medical University, Taipei City 11031, Taiwan

**Keywords:** prostate cancer, HSD17B2, HSD17B3, SHBG and SRD5A1, advanced disease, castration-resistance, therapy resistance, testosterone metabolism, androgenic reprogramming

## Abstract

**Simple Summary:**

Over the last two decades, our improved understanding of the pathobiology of androgen-addicted prostate cancer (PCa), and documented therapeutic advances/breakthroughs have not translated into any substantial or sustained curative benefit for patients treated with traditional ADT or novel immune checkpoint blockade therapeutics. This is invariably connected with the peculiar biology and intratumoral heterogeneity of PCa. Castration-resistant PCa, constituting ~30% of all PCa, remains a clinically enigmatic and therapeutically challenging disease sub-type, that is therapy-refractory and characterized by high risk for recurrence after initial response. Our findings highlight the role and exploitability of testosterone metabolic reprogramming in prostate TME for patient stratification and personalized/precision medicine based on the differential but concerted expression of molecular components of the proposed testosterone tetrad in patients with therapy-refractory, locally advanced, or recurrent PCa. The therapeutic exploitability and clinical feasibility of our proposed approach is suggested by our preclinical findings.

**Abstract:**

*Background*: Testosterone plays a critical role in prostate development and pathology. However, the impact of the molecular interplay between testosterone-associated genes on therapy response and susceptibility to disease relapse in PCa patients remains underexplored. *Objective*: This study investigated the role of dysregulated or aberrantly expressed testosterone-associated genes in the enhanced dissemination, phenoconversion, and therapy response of treatment-resistant advanced or recurrent PCa. *Methods:* Employing a combination of multi-omics big data analyses, in vitro, ex vivo, and in vivo assays, we assessed the probable roles of HSD17B2, HSD17B3, SHBG, and SRD5A1-mediated testosterone metabolism in the progression, therapy response, and prognosis of advanced or castration-resistant PCa (CRPC). *Results:* Our bioinformatics-aided gene expression profiling and immunohistochemical staining showed that the aberrant expression of the HSD17B2, HSD17B3, SHBG, and SRD5A1 testosterone metabolic tetrad characterize androgen-driven PCa and is associated with disease progression. Reanalysis of the TCGA PRAD cohort (n = 497) showed that patients with SRD5A1-dominant high expression of the tetrad exhibited worse mid-term to long-term (≥5 years) overall survival, with a profoundly shorter time to recurrence, compared to those with low expression. More so, we observed a strong association between enhanced HSD17B2/SRD5A1 signaling and metastasis to distant lymph nodes (M1a) and bones (M1b), while upregulated HSD17B3/SHBG signaling correlated more with negative metastasis (M0) status. Interestingly, increased SHBG/SRD5A1 ratio was associated with metastasis to distant organs (M1c), while elevated SRD5A1/SHBG ratio was associated with positive biochemical recurrence (BCR) status, and shorter time to BCR. Molecular enrichment and protein–protein connectivity network analyses showed that the androgenic tetrad regulates testosterone metabolism and cross-talks with modulators of drug response, effectors of cell cycle progression, proliferation or cell motility, and activators/mediators of cancer stemness. Moreover, of clinical relevance, SHBG ectopic expression (SHBG_OE) or SRD5A1 knockout (sgSRD5A1) induced the acquisition of spindle fibroblastoid morphology by the round/polygonal metastatic PC-3 and LNCaP cells, attenuated their migration and invasion capability, and significantly suppressed their ability to form primary or secondary tumorspheres, with concomitant downregulation of stemness KLF4, OCT3/4, and drug resistance ABCC1, ABCB1 proteins expression levels. We also showed that metronomic dutasteride synergistically enhanced the anticancer effect of low-dose docetaxel, in vitro, and in vivo. *Conclusion:* These data provide proof of concept that re-reprogramming of testosterone metabolism through “SRD5A1 withdrawal” or “SHBG induction” is a workable therapeutic strategy for shutting down androgen-driven oncogenic signals, reversing treatment resistance, and repressing the metastatic/recurrent phenotypes of patients with PCa.

## 1. Introduction

Prostate cancer (PCa) is among the most diagnosed male malignancies globally. Considering that ~1/3 of patients who undergo radical prostatectomy for clinically localized PCa suffer postoperative recurrence, androgen deprivation therapy (ADT) remains the treatment of choice for advanced and clinically localized PCa. Identifying patients at high risk of metastasis or recurrence after prostatectomy can inform medical management and improve prognosis. The last decade has been characterized by increased exploration of selected genes and proteins (so called proteogenomics) in the oncogenicity, immunogenicity, disease progression, and therapy response, with piqued interest in how this gene(s) dictates treatment success or failure in patients (otherwise called precision medicine). This seamless integration of proteogenomics and precision medicine (herein termed “precision proteogenomics”) is touted to provide comprehensive elucidation of disease mechanisms through discovery and validation of novel diagnostic/prognostic biomarkers, while concomitantly facilitating patient stratification into responders or non-responders to specific targeted therapies.

### 1.1. Clinical and Molecular Characteristics of Prostate Cancer

Prostate cancer (PCa) is the most diagnosed non-cutaneous male malignancy and a leading cause of cancer-related deaths globally, having an annual incidence and mortality rate of 7.1 and 3.8%, respectively, in 2018, and a projected increase of ~1.8-fold in incidence and 2.1-fold in mortality by the year 2040 [1]. It has been suggested that one in every three patients with PCa will develop metastatic or recurrent disease within 2 years of initial diagnosis, one in every five patients with metastatic or recurrent disease evolve into castration-resistant PCa (CRPC) by fifth year of follow-up, and the median survival of patients after developing castration resistance is ~14 months [2,3]. Despite this soaring incidence, high mortality burden, and enhanced risk of disease progression among patients with PCa, the biomechanisms underlying its development and progression remain largely unclear. In the last decade, there has been a renaissance and increased exploration of the role of androgens, including testosterone and its derivative, dihydrotestosterone (DHT), in the cancerization of normal prostate cells, disease progression, and poor prognosis in patients with PCa; this may not be unconnected with the androgen-regulated nature of the prostate, as well as the indispensability of androgens in the regulation of uptake and metabolism of lipid and insulin [4]. Against this background, PCa remains a profoundly heterogeneous and complex pathology, such that the clinicopathological characteristics of patients at diagnosis fails to adequately anticipate tumor clinical phenotype, and needle biopsy-associated undersampling undermines the reliability of biopsy findings as truly reflective of disease course or cancer aggressiveness. All these necessitate the discovery and validation of actionable molecular biomarkers that provide clinically objective measures of the tumor biology, improve the patients’ classification and stratification, and inform therapeutic decision-making (see Graphic Abstract).

### 1.2. Testosterone Metabolism, Therapy Response, and Susceptibility to Disease Relapse in Patients with PCa

Despite diagnostic and therapeutic advances made in the last 3 decades, PCa morbidity remains high, about 30% of patients present with advanced or metastatic disease, and mortality is unabating. Following Huggins and Hodges’ concept of androgen-dependence introduced in 1972, androgen deprivation therapy (ADT) has remained the standard of care for initial management of advanced, metastatic, or recurrent disease; however, development of CRPC within 2–3 years of ADT initiation continues to confound the promise of any therapeutic success in prostate cancer clinics [2,3]. A principal characteristic of PCa cells is their profound dependence on or addiction to androgens. Androgens are essential for prostate physiology in men. As already alluded, the principal male androgens are testosterone, produced by testicular interstitial or Leydig cells, and DHT, derived from steroid-5-α-reductase (SRD5A1)-mediated catabolism of testosterone in peripheral tissues. SRD5A messenger RNA expression profiling in several publicly available cancer datasets show that, compared with nontumor prostate tissues, SRD5A1 is upregulated in primary and metastatic PCa, and the activation of the driver oncogene androgen receptor (AR) induced a 2- to 4-fold overexpression of SRD5A1 in androgen-responsive PCa cells [5].

In circulation, testosterone is primarily bound to sex-hormone-binding globulin (SHBG), while the free or unbound testosterone is the active and most bioavailable form [6]. The internalization and endogenous expression of SHBG has been shown to increase DHT uptake, prolong the expression of testosterone-responsive genes, and is associated with clinicopathological traits that are characteristic of disease progression. In fact, it has been suggested that reduced or absent serum SHBG, is associated with enhanced testosterone uptake, glucuronidation, and efflux, thus, increasing testosterone deficiency and driving cancerous cell proliferation and disease progression [7]. More so, ~54.3% of patients with metastatic AR positive tumors concurrently expressed enzymes for adrenal androgen utilization, such as SRD5A1 and SHBG, and 25.7% expressed enzymes for de novo steroidogenesis, including hydroxysteroid 17-β-dehydrogenase 2 and 3 (HSD17B2/3) [8]. It has been suggested that HSD17B2 suppresses androgen production by reverse conversion of testosterone or DHT to their upstream precursors, that the expression of HSD17B2 reduces as PCa progresses, and that the overexpression of HSD17B2 suppresses androgen-induced cell proliferation and xenograft growth [9].

### 1.3. The Clinical Implication of Dysregulated Testosterone Metabolism in Therapy Response and Susceptibility to Disease Relapse in Patients with PCa

Predicting therapy response and/or clinical outcome in patients with newly diagnosed PCa is challenging. This is in part because of the current non-standardized imaging methods for assessing disease dissemination and the confounding dynamism of the most frequently altered PCa-associated biomarker, prostate-specific antigen (PSA), which makes the latter a less reliable or accurate surrogate biomarker of disease course or treatment response. For instance, about 20% of patients with CRPC who later respond to chemotherapy would have been tagged “non-responders” because of an initial persistent rise in PSA level, which did not decline until after week 12 of chemotherapy or did not decline at all when on immune checkpoint blockade therapy [10]. This modest or non-association between changes in post-treatment PSA level and therapy response or disease recurrence highlights a critical unmet need in PCa management—the need for more reliable and accurate indicators of patient status, namely therapy response or disease recurrence.

Advances in tumor biology increasingly highlight the genomic complexity of cancerous cells, irrespective of tissue origin or histological sub-type [11]. However, within this broad genomic/genetic landscape, some cancer types are more dependent on certain oncogenic pathways for survival than others. This state of preferential “oncogene addiction” is common with aberrant oncometabolic activity, and provides therapeutic basis for molecular targeting of dysregulated oncogenic metabolites [11]. The apparent dependence of cancerous prostate cells on androgen/testosterone metabolic signaling for their survival and the maintenance of their malignant therapy-resistant and recurrent phenotypes makes molecular components of testosterone metabolic reprogramming exploitable for reliably accurate prediction of disease course, therapy response, and clinical outcome, thus, aiding patient stratification and informing therapeutic decision-making when managing patients just diagnosed with PCa.

The present study harnesses the profiling of disease-relevant molecular players, namely testosterone metabolites, HSD17B2, HSD17B3, SHBG, and SRD5A1 to provide an evidence-based platform for exploring and identifying biomarkers that may inform patient stratification, allow prediction of treatment efficacy, and determine mechanism(s) of drug resistance. This, we believe, would serve as a putative basis for precision medicine/ individualized therapy for patients with PCa.

### 1.4. The Complicity of Testosterone-Addiction and Testosterone Metabolic Reprogramming in the Response of PCa Cells to Therapy, and Propensity for Recurrence

The last two decades have been characterized by enthralling evidence from several studies indicating that the survival of cancerous cells depends on selected principal “driver” genomic events, and this bio-phenomenon of intricate dependence of cancerous cells on certain oncogenes to sustain their malignant phenotype is herein termed “oncogene addiction” [10]. In this study, we hypothesized that, within the context of testosterone dynamism, PCa cells are addicted to testosterone and its metabolites, that the malignant and therapy-resistant phenotype of PCa cells are, to a large extent, dependent on the well-knit molecular interplay between components of the testosterone metabolism, and that metabolic normalization of dysregulated testosterone is attainable by restoration of aberrantly expressed and/or constitutively suppressed molecular effectors of testosterone metabolic reprogramming to physiologic regulation.

Understanding that genetic/genomic manipulation of one or more testosterone-associated molecular players in experimental models may be sufficient to cause tumor recurrence in vivo, we posit that deferentially exploiting the addiction of PCa cells to the testosterone metabolites, HSD17B2, HSD17B3, SHBG, and SRD5A1 induces probable differentiation and/or loss of self-replication of the aggressive PCa cells, and elicits post-therapy complete response or long-term remission. This is clinically relevant in the context of well-documented unabating incidence of recurrent PCa after primary curative therapy, with a 19 to 35% incidence of biochemical recurrence (BCR) at 10 years post-radical prostatectomy and ~30% post-radiotherapy [12].

Molecular fine tuning of the PCa genomic landscape by reprogramming of testosterone metabolites, HSD17B2, HSD17B3, SHBG, and SRD5A1 holds the promise of profoundly altering the tumor microenvironment (TME), suppressing angiogenesis, and enhancing susceptibility of the cancerous cells to immunosurveillance [13,14,15]. Recent reports indicate that “metastatic prostate cancer initially retains its androgen dependence, and androgen-deprivation therapy often leads to disease control; however, the cancer inevitably progresses despite treatment” and that “these tumors evade treatment via mechanisms that augment acquisition of androgens from circulating precursors, increase sensitivity to androgens and androgen precursors, bypass the androgen receptor, or a combination of these mechanisms” [16]. Thus, we tested the hypothesis that the concerted molecular interplay between testosterone-associated HSD17B2, HSD17B3, SHBG, and SRD5A1 determine therapy response and susceptibility to disease relapse in patients with PCa. The study provides proof of concept that dysregulated or aberrantly expressed HSD17B2, HSD17B3, SHBG, and SRD5A1 drive enhanced dissemination, phenoconversion, and therapy response of aggressive metastatic, treatment-resistant (platinum-based chemotherapy, ADT and/or immunotherapy) unresectable/advanced or recurrent PCa cells.

### 1.5. Translational Relevance of Present Study

It is common knowledge (i) that the prostate is an androgen-addicted organ, (ii) that recurrent PCa, presenting as localized recurrence and/or distant metastatic disease, retains androgen dependence, and (iii) that PCa evades therapy using mechanisms that amplify testosterone uptake, increase sensitivity to testosterone or its metabolites, circumvent AR, or combine the three mechanisms. However, the exploitation of “testosterone metabolism reprogramming” as a veritable tool for patient stratification or prediction of therapy response and susceptibility to disease recurrence remains largely unexplored. Based on this paradigm, this study highlights the critical role of the concerted interplay between testosterone metabolites HSD17B2, HSD17B3, SHBG, and SRD5A1 as reflective of disease status and as a surrogate biomarker of therapy response in patients with PCa. This study lays the groundwork for future multicenter large cohort clinical trial on the predictive accuracy of the testosterone tetrad. Preclinical data presented herein suggest the clinical feasibility of the testosterone tetrad as indicators of patient status and as reliable predictors of therapy response and susceptibility to disease recurrence.

## 2. Material and Methods

### 2.1. Prostate Cancer Tissue Samples

We obtained prostate cancer tissue samples (n = 56) from the Taipei Medical University–Shuang Ho Hospital tissue bank, following ethical approval for their use from the Institutional Review Board of the Taipei Medical University (approval number: N202101071). The requirement for patients’ signed informed consent was waived because the tissue samples were obtained retrospectively from the Taipei Medical University–Shuang Ho Hospital PCa archive.

### 2.2. Cell Culture and Chemicals

Normal human primary prostate epithelial HPrEC (ATCC^®^ PCS-440-010™), PSA-secreting primary prostate carcinoma 22Rv1 (ATCC^®^ CRL-2505™), 5-a dihydrotestosterone -responsive, androgen-dependent metastatic prostate carcinoma LNCaP (ATCC^®^ CRL-1740™), and low acid phosphatase and testosterone-5-a reductase, androgen-independent metastatic prostate carcinoma PC-3 (ATCC^®^ CRL-1435™) cell lines were obtained from the ATCC (American Type Culture Collection, Manassas, VA, USA) and cultured in RPMI1640 (Thermo Fisher Scientific Inc., Bartlesville, OK, USA). Culture medium was supplemented with 10% fetal bovine serum (FBS, #26140079, Thermo Fisher Scientific Inc., Bartlesville, OK, USA) and 100 U/mL of penicillin–streptomycin (Thermo Fisher Scientific Inc., Bartlesville, OK, USA). All cells used in the study were not greater than passage number 2 (≤P.2). Cells were sub-cultured at ≥95% confluence and culture media changed every 48 h. Stock solutions of 100 mM in 0.01% DMSO were stored at −20 °C until use.

### 2.3. Antibodies and Reagents

Monoclonal antibodies against HSD17B2 (#TA504616), HSD17B3 (#CF811500), SHBG (#TA507187), and SRD5A1 (#PA5-75675) were obtained from Thermo Fisher Scientific Inc. (Bartlesville, OK, USA), while KLF4 (#12173), OCT4A (#2840), MRP1/ABCC1 (#14685), MDR1/ABCB1 (#13342), and ALDH1 (#36671) were all purchased from Cell Signaling Technology (CST, Beverly, MA, USA), and GAPDH (#sc-32233) was from Santa Cruz Biotechnology (Santa Cruz, CA, USA). Dutasteride (#SML1221, ≥98% (HPLC)) was purchased from Sigma-Aldrich Inc. (St. Louis, MO, USA).

### 2.4. Cell Viability and Proliferation Colorimetric Assay

Cell viability was assessed using the sulforhodamine B (SRB) assay. A total of 3 × 10^3^ wild type (WT) or SHBG-overexpressing (SHBG_OE) prostate cancer cells were seeded per well in 96-well microtiter plates containing supplemented growth media, and incubated at 37 °C in humidified 5% CO_2_. After 48 h, cell viability was measured following the manufacturer’s instructions. After fixing the WT or SHBG_OE prostate cancer cells with 10% trichloroacetic acid (TCA) and carefully washing with ddH_2_O, the cells were stained with 0.4:1 (*w/v*) SRB–acetic acid solution. We carefully washed off unbound SRB dye from the cells using 1% acetic acid thrice, followed by air-drying the plates, and solubilization of bound SRB dye in 10 mM Tris base. For cell proliferation, invitrogen alamarBlue™ high sensitivity cell viability reagent (#A50100, Thermo Fisher Scientific Inc., Bartlesville, OK, USA) was used, strictly following the manufacturer’s instructions. Briefly, after seeding cells in triplicates with three biological replicas for each assay at each time point (day 1–8), the cells were incubated with alamarBlue™ for 2 h at 37 °C. The number of dye-stained viable/proliferating cells was read at 570 nm absorbance wavelength in the Molecular Devices Spectramax M3 multimode microplate reader (Molecular Devices LLC., San Jose, CA, USA).

### 2.5. Drug Combination Assay

The synergistic effect of Docetaxel, combined with or without Dutasteride, was evaluated by adapting the Chou–Talalay algorithm for multidrug combinations. Combined drug interactions were evaluated by isobologram and combination index (CI) values derived using CompusSyn (CompuSyn, Inc., Paramus, NJ, USA). A CI of <1, 1, and >1 represented drugs’ synergism, additivity, and antagonism, respectively. When all combination dose-points lay within the right-angled “isobologram” triangle, synergism was inferred; by contrast, dose-points laying on the hypotenuse indicated additivity, and, when the dose-points fell outside the isobologram, the combined drugs were designated antagonistic. We also used the SynergyFinder web application version 2.0 (https://synergyfinder.fimm.fi/, accessed on 26 January 2021) for validation of observed multidrug synergism.

### 2.6. Knockout of SRD5A1 by CRISPR Interference

Plasmid vectors containing pLenti-U6-sgRNA-SFFV-Cas9-2A-Puro (#454421110502; SRD5A1 sgRNA CRISPR All-in-One Lentivirus set (Human); Applied Biological Materials Inc., Richmond, BC, Canada) was used to knockout SRD5A1 in cells by CRISPR interference. Three SRD5A1-specific single-guide RNAs (sgRNAs) designed using the genetic perturbation platform (GPP) sgRNA Designer tool were synthesized and separately cloned into lenti-dCas9-2A. Lentiviruses were packaged and transfected into PC-3 cells. Stably transfected monoclonal PC-3 cells were selected by 2 μg/mL puromycin, as recommended by the manufacturer. The knockout of SRD5A1 in cells was verified by genomic sequencing and quantitative real-time PCR. The sgRNA sequences for SRD5A1 are as follows: sgSRD5A1#1 5′-GTCCCGGCAGTGCGGGACTC-3′, sgSRD5A1#2 5′-GACTCCGGTAGCCGCCCCTC-3′, and sgSRD5A1#3 5′-GCTACCGGAGGGGCGGCTAC-3′.

### 2.7. Construction and Transfection of Plasmids Expressing SHBG

Ectopic expression of SHBG was achieved using the sex hormone binding globulin (SHBG) (NM_001040) Human Untagged Clone (#SC302958, OriGene Technologies, Inc., Rockville, MD, USA) in pCMV6-XL5 vectors transfected into PC-3 cells using Lipofectamine 2000 reagent (Invitrogen, Carlsbad, CA, USA) according to the manufacturer’s instructions. Cells transfected with empty vector served as controls. Clones stably expressing SHBG were selected by 100 mg/mL ampicillin (#11593027, Thermo Fisher Scientific Inc., Bartlesville, OK, USA).

### 2.8. Immunohistochemical (IHC) and Immunofluorescence (IFC) Staining Assays

Immunohistochemical (IHC) analyses were performed on formalin-fixed paraffin-embedded (FFPE) sections from our PCa cohort consisting of patients with different tumor grades (normal: Gleason score (GS) ≤ 5; low: GS = 6–7; high: GS ≥ 8) and therapy response. The study was approved by the Taipei Medical University Institutional Review Board (approval number: N202101071) and compliant with recommendations from the Declaration of Helsinki for biomedical research involving human subjects. Samples were probed with antibodies against HSD17B2, HSD17B3, SHBG, SRD5A1, KLF4, OCT4A, MRP1, MDR1, ALDH1, and GAPDH at 1:200 dilution following standard IHC protocol. Protein expression was scored by two independent pathologists using the quick-score (Q-score) formula Q = I × P, where I is staining intensity (0 (no staining), 1+ (weak), 2+ (moderate), and 3+ (strong)), and P represents percentage of stained cells. Maximum Q-score = 300. For immunofluorescence (IFC) staining, WT, sgSRD5A1, or SHBG_OE PCa cells or tumorspheres derived from the PCa cells were plated onto poly-*L*-lysine-coated glass cover-slips, fixed with 4% paraformaldehyde, washed carefully with cold PBS thrice, permeabilized with 0.1% Triton X-100/PBS solution for 10 min, and then incubated with primary antibodies, followed by Cy5-labeled goat anti-mouse Alexa Fluor488 secondary antibodies (#R37120, Thermo Fisher Scientific Inc.) for 1 h. DAPI (4′,6-diamidino-2-phenylindole; #D1306, Molecular Probes, Thermo Fisher Scientific Inc.) was used for nuclear staining. For cell visualization and imaging, the Nikon E800 fluorescent microscope (Nikon Instruments Inc., Melville, NY, USA) was used.

### 2.9. Western Blotting Assay

After separating 20 µg of protein samples from WT, sgSRD5A1, or SHBG_OE PCa cells using 10% sodium dodecyl sulfate polyacrylamide gel electrophoresis (SDS–PAGE), protein blots were transferred onto polyvinylidene fluoride (PVDF) membranes using the Bio-Rad Mini-Protein electro-transfer system (Bio-Rad Laboratories, Inc., Hercules, CA, USA). The PVDF membranes were then blocked with 5% skimmed milk in Tris-buffered saline with Tween 20 (TBST) for 1 h, followed by incubation overnight at 4 °C with primary monoclonal antibodies against HSD17B2 (1:2000, ThermoFisher Scientific), HSD17B3 (1:2000, ThermoFisher Scientific), SHBG (1:1000, ThermoFisher Scientific), SRD5A1 (1:1000, ThermoFisher Scientific), KLF4 (1:1000, Cell Signaling Technology), OCT4A (1:1000, Cell Signaling Technology), MRP1 (1:1000, Cell Signaling Technology), MDR1 (1:1000, Cell Signaling Technology), ALDH1 (1:1000, Cell Signaling Technology), and GAPDH (1:1000, Santa Cruz Biotechnology). Thereafter, the membranes were incubated with secondary antibodies conjugated with horseradish peroxidase (HRP) for 1 h at room temperature, washed carefully three times with cold 1X PBS, and then protein bands were detected with the enhanced chemiluminescence detection system (Thermo Fisher Scientific Inc., Waltham, MA, USA), and protein band densitometry was done using ImageJ software version 1.49 (https://imagej.nih.gov/ij/, accessed on 14 December 2020).

### 2.10. Tumorsphere Formation and Self-Renewal Assay

A total of 5 × 10^4^ WT, sgSRD5A1, or SHBG_OE PC-3 and LNCaP cells were seeded per well in ultra-low attachment 6-well plates (Corning, Corning, NY, USA) containing RPMI1640 supplemented with 20 ng/mL basic fibroblast growth factor (bFGF; #13256029, Invitrogen), GibcoTM B-27TM supplement (#17504044, Invitrogen, Carlsbad, CA, USA), and 20 ng/mL epidermal growth factor (EGF; #PHG0311, Invitrogen). Cells were cultured at 37 °C in a humidified 5% CO_2_ incubator for 5–7 days. Formed primary tumorspheres ≥ 100 µm were counted under an inverted phase-contrast microscope. Furthermore, secondary tumorspheres were generated by dissociating the primary tumorspheres, and reseeding cells, as per the primary tumorspheres, from single-cell suspension acquired using a sterile 22G needle.

### 2.11. Scratch-Wound Healing Migration Assay

For cell migration, we used the scratch wound-healing assay. Briefly, WT or SHBG_OE PCa cells were seeded and allowed to grow in 6-well plates (Corning, Corning, NY, USA) containing complete growth media with 10% FBS. Media in wells were changed to low serum (1% FBS) growth media when cells attained >98% confluence. The median axes of the mono-layered adherent cells were denuded using sterile yellow pipette tips. Cell migration based on scratch-wound healing was monitored over time, and images were captured at 0 and 24 h after denudation under a light microscope using a 10X objective lens. Thereafter, the images were analyzed using National Institutes of Health ImageJ software version 1.49 (https://imagej.nih.gov/ij/, accessed on 20 December 2020).

### 2.12. Invasion Assay

Invasion assay was performed using the Corning^®^ BioCoat™ Matrigel^®^ Invasion Chambers with 8.0 μm PET membrane in two 24-well plate systems (#354480, Corning, Corning, NY, USA). A total of 1 × 10^5^ WT or SHBG_OE PCa cells were seeded per well in plates and incubated at 4 °C overnight. The upper chambers contained low serum (2% FBS) media, while the lower chamber contained 600 μL high serum (20% FCS) media. After 48 h incubation, the noninvaded cells in the upper chamber were carefully wiped off with sterile cotton swabs, while the invaded cells that penetrated through the membrane were fixed with ethanol, stained with crystal violet solution, and counted under a light microscope from six random fields of vision.

### 2.13. Tumor Xenograft In Vivo Studies

For in vivo tumor xenograft studies, 1 × 10^6^ PC-3_WT or PC-3_SHBG_OE cells in 100 mL complete growth medium were injected into the left cardiac ventricle of 7–8-week-old male BALB/c-nu mice (28.3 ± 5.2 g; n = 10 per treatment group) (BioLASCO, Taipei City, Taiwan), subcutaneously. Mice were randomly placed into control (PC-3_WT) or test (PC-3_SHBG_OE, PC-3_sgSRD5A1, PC-3_WT + Docetaxel + Dutesteride) groups. For the drug treatment group, treatment was initiated as soon as the tumors became palpable (tumor volume ~105 mm^3^). A 100 mL/day vehicle 0.01% DMSO was given intraperitoneally (i.p.) to the control mice inoculated with PC-3_WT. Docetaxel 10 mg/kg/day and Dutesteride 2.5 mg/kg/day i.p. was given every 72 h for 8 weeks. Tumor growth was monitored throughout the experiment, with tumors measured with calipers twice a week and volume estimated using the formula: 0.5 (length (mm)) × (width (mm))^2^. The mice were humanely sacrificed when tumor size became nonsurvivable, or on day 80. The tumors were then excised and carefully analyzed. Metastatic nodules in extracted lungs, livers, brain, and bone were also assessed. Animal studies complied with approved protocol of the Lab Animal Committee/Institutional Animal Care and Use Committee (Approval no.: LAC-2020-0553) of Taipei Medical University.

### 2.14. Statistical Analysis

All data represent the mean ± standard deviation (SD) of assays performed at least 3 times in triplicates. The 2-sided Student’s *t* test was used for comparison between 2 groups, whereas one-way ANOVA with Tukey’s post hoc test was used for comparison between 3 or more groups. Kaplan–Meier survival analyses aided comparison of survival rates between the control and test group. All statistical analyses were performed using GraphPad Prism version 8.0.0 for Windows (GraphPad Software, La Jolla, CA, USA). *p*-value < 0.05 was considered statistically significant.

## 3. Results

### 3.1. The Aberrant Expression of the Testosterone Metabolic Tetrad HSD17B2, HSD17B3, SHBG, and SRD5A1 Characterize Androgen-Addicted PCa and Is Associated with Disease Progression

Computational analysis of the gene ontology (GO) biological process complete dataset from the GO Ontology database version 8 December 2020 (http://doi.org/10.5281/zenodo.4316524, accessed on 3 November 2020) using the Fisher’s exact over-representation test indicate that HSD17B2, HSD17B3, SHBG, and SRD5A1 are significantly enriched for androgen biosynthetic processes (fold enrichment > 100, *p* = 1.82 × 10^−6^) and male genitalia development (fold enrichment > 100, *p* = 8.27 × 10^−6^) (Figure 1A). Bioinformatics-aided gene expression profiling showed that, compared to HSD17B2 (0.31-fold, *p* = 0.0092) and SHBG (0.93-fold, *p* = 0.78) mRNA expression levels, which are downregulated, HSD17B3 (2.96-fold, *p* = 1.11 × 10^−16^) and SRD5A1 (1.3-fold, *p* = 4.02 × 10^−5^) transcript levels are upregulated in patients with PCa from the TCGA PRAD cohort (n = 497) (Figure 1B). Furthermore, using samples from our PCa cohort (n = 56), compared to the moderate-strong immunoreactivity of HSD17B3 and SRD5A1 in high-grade PCa (Gleason score (GS) ≥ 8), normal prostate tissue (GS ≤ 5), and low-grade PCa (GS = 6–7) were characterized by null-mild protein expression, conversely, moderate-strong HSD17B2 and SHBG protein expression levels were observed in the normal prostate and low grade PCa tissues compared to null–mild protein expression in the high-grade samples (Figure 1C). Consistent with this, reanalysis of the TCGA PRAD cohort (n = 497) showed that tumor T-stage progression was associated with SRD5A1 and HSD17B3 gene amplification and gain, but shallow or deep deletion of SHBG and HSD17B2 genes (Figure 1D). The differential expression of HSD17B2, HSD17B3, SHBG, and SRD5A1 in patients with different PCa grades is suggestive of a dysregulated androgenic signal-driven oncogenicity in patients with PCa, where intracellular signaling skewed towards a dominant SRD5A1/HSD17B3 at the expense of SHBG/HSD17B2 signaling drives enhanced cancerization, and may inform therapeutic decision making and management strategy for patients with PCa.

### 3.2. Variation in HSD17B2, HSD17B3, SHBG, and SRD5A1 Expression Co-Occur and Concertedly Bear Significant Prognostic Relevance in Patients with PCa

Ruling out random individualistic expression pattern, our computational mutual exclusivity and co-occurrence probe of a pooled PCa cohort (n = 4369) showed that all four molecular components of the testosterone metabolic tetrad co-occur, and the co-occurrence is statistically significant (Figure 2A). Since molecular co-occurrence is often suggestive of functional inter-relatedness, we examined if and to what degree the tetrad is culpable in risk of death and recurrence in patients with PCa. We observed that, while HSD17B2, HSD17B3, and SHBG expression levels were equivocal for risk of death, high expression of SRD5A1 was strongly associated with high risk of death (Figure 2B,C). However, all four components of the tetrad were fully active in driving the propensity for relapse, with concomitant high expression of SRD5A1 and low HSD17B2, HSD17B3, and SHBG being associated with significantly enhanced risk of recurrence (Figure 2D,E). In corroboration, Kaplan–Meier curves were generated from survival analyses of HSD17B2/HSD17B3/SHBG/SRD5A1 co-expression in TCGA PRAD cohort (n = 497), showing that, compared to the low expression group, patients with SRD5A1-dominant high tetrad expression exhibited worse mid-term to long-term (≥5 years) overall survival ((HR = 5.19 (95% CI: 1.09–24.63); *p* = 0.04) (Figure 2F), and a profoundly shorter time to recurrence ((HR = 3.6 (95% CI: 1.59–8.12); *p* = 0.002) (Figure 2G). The group expression cutoff was based on 75% (high)/25% (low) quartile. These data demonstrate that the concerted (but not individual) activity of the tetrad exhibited a high prognostic index, indicating that the differential expression pattern of the HSD17B2/HSD17B3/SHBG/SRD5A1 tetrad in androgen-addicted PCa has prognostic implications relevant for prediction of the clinical outcome.

### 3.3. The Differential but Concerted Expression of HSD17B2, HSD17B3, SHBG, and SRD5A1 Is Associated with the Metastatic and Recurrent Phenotype of Patients with PCa

Because of the interplay between disease aggression, progression, and prognosis, we statistically reanalyzed the National Cancer Institute Genomic Data Commons (NCI GDC) TCGA PRAD cohort (n = 623) for probable correlation between the differential expression of the testosterone tetrad and PCa metastasis or recurrence. We observed strong association between high HSD17B2/SRD5A1 expression and metastasis to distant lymph nodes (M1a) and bones (M1b) in patients with PCa, while high HSD17B3/SHBG expression correlated more with negative metastasis status (M0) (Figure 3A). Interestingly, both low SRD5A1 and high SHBG expressions were associated with metastasis to distant organs (M1c) (Figure 3A). Moreover, low SHBG or HSD17B2 and high SRD5A1 or HSD17B3 expression levels were associated with positive biochemical recurrence (BCR) status (Figure 3B), and shorter time to BCR (Figure 3C). Also of clinical relevance, we observed that, while high SHBG and low SRD5A1 were associated with complete or partial response (CR/PR), the opposite was true for progressive disease (PD) (Figure 3D). Furthermore, IHC analysis of our PCa cohort (n = 56) showed that, compared to the nontumor samples, SRD5A1 and HSD17B3 protein expression levels were significantly upregulated in patients with primary and metastatic PCa, but SHBG and HSD17B2 protein expression levels were markedly suppressed (Figure 3E). These data highlight the nonrandom, but concerted and well-calibrated interplay between the four androgenic facilitators of disease progression and determinants of therapy response, as well as suggest disease driving roles for SHBG and SRD5A1, while HSD17B2 and HSD17B3 are complicit passengers.

### 3.4. Distinct Interaction between the Testosterone Tetrad Elicits Androgenic Signals That Drive Cell Cycle Progression, Enhanced Motility, Cancer Stemness, and Resistance to Therapy in Patients with PCa

To determine the molecular linkage between SRD5A1/SHBG/HSD17B2/HSD17B3 androgenic signaling and therapy response, we performed molecular connectivity and functional enrichment analyses using the STRING software version 11.0 (https://string-db.org/, accessed on 11 February 2021). Our bioinformatics-aided visualization shows a significantly high protein–protein interaction (PPI) enrichment between the tetrad (*p* = 6.43 × 10^−9^), and suggests an active role for SRD5A1 in the transcriptional regulation of SHBG, the HSD17B2/HSD17B3 complex (Figure 4A), and cytochrome P450 family 17 subfamily A member 1 (CYP17A1) (Figure 4B). This indicates, at least in part, a critical role for the testosterone tetrad in drug metabolism, coupled with steroids, cholesterol, and lipid biosynthesis—a dyad that modulates cancerization, disease progression, and treatment success. Furthermore, molecular docking shows that the HSD17B2/HSD17B3 complex (docking score = −359.46; ligand root-mean-square deviation, RMSD = 49.54 Å) directly interacts with the SRD5A1/SHBG complex (docking score = −296.02; ligand RMSD = 69.47 Å) to form a HSD17B2/HSD17B3/SHBG/SRD5A1 macro-complex (docking score = −355.07; ligand RMSD = 114.28 Å), posited herein to drive the androgenic cum recurrent phenotype of patients with PCa (Figure 4C). Furthermore, AFFY_HG_U133_PLUS_2 expression profiling array analysis of the stromal molecular signatures of prostate and breast cancer using the GSE26910 dataset (n = 24) showed that, compared to downregulated SHBG, HSD17B2 and HSD17B3 expression, upregulated expression of SRD5A1 in tumor samples positively correlated with upregulation of ATP-binding cassette subfamily C member 1/multidrug resistance-associated protein 1 (ABCC1/MRP1), multidrug resistance protein 1 (ABCB1/MDR1) (Figure 4D), NANOG, SOX2, and POU Class 5 Homeobox 1 (POU5F1/OCT4) (Figure 4E). Reanalysis of the GSE33455 expression data from docetaxel-resistant PCa cell lines (n = 12) showed that, compared to the WT cells, SRD5A1, ALDH1A1, ABCC1, and ABCB1 expression were concomitantly upregulated in docetaxel-resistant DU-145 cells, while SHBG, HSD17B2, HSD17B3, and ABCB1 were enhanced in docetaxel-resistant PC3 cells (Figure 4F). More so, using the GSE3325 dataset (n = 19), we observed markedly upregulated SRD5A1, HSD17B2, POU5F1, SOX2, and NANOG in the metastatic PCa, and HSD17B3, CD44, POU5F1, ABCB1, prominin 1 (PROM1/CD133), ABCC1, NANOG, and Kruppel-like factor 4 (KLF4) in primary PCa (Figure 4G). In concordance, automated molecular enrichment and analyses of our protein–protein connectivity network indicate that the SHBG/HSD17B2/HSD17B3/SRD5A1 androgenic signaling regulates testosterone metabolism and cross-talks with effectors of drug response, including CYP17A1, ABCB1, ABCC1, ABCG2, and ALDH1A1; modulators of cell cycle progression, proliferation, and cell motility, such as cyclin-dependent kinase (CDK)1, targeting protein for Xklp (TPX)2, aurora kinase A (AURKA), and hyaluronan-mediated motility receptor (HMMR); and activators/regulators of cancer stemness, including POU5F1, SOX2, LIN28A, KLF4, and NANOG (Figure 4H). The data indicate that the direct interaction between HSD17B2, HSD17B3, SHBG, and SRD5A1 is not just spatiotemporal, but functionally primed for therapy response, and may be therapeutically actionable.

### 3.5. Molecular Fine-Tuning of Components of the 4-Gene PCa Signature Modulate the Highly Proliferative, Metastatic and Cancer Stem-Cell-Like (Cum Disease Recurrent) Phenotypes of PCa Cells

Using the Cancer Cell Line Encyclopedia (CCLE) RNA-seq data of human prostate cancer cell lines, 22Rv1, DU145, LNCaP, MDA PCa 2b, NCI-H660, PC-3, and VCaP, we observed that, compared with SHBG, which is endogenously not expressed, SRD5A1 is significantly upregulated in recurrent and metastatic PCa cell lines (Figure 5A). To provide some mechanistic insight, we demonstrated that the ectopic expression of SHBG (SHBG_OE) inhibited the proliferation (day 8: 83.7% reduction vs. WT, *p* < 0.001) (Figure 5B), suppressed the viability (day 8: 74.5% reduction vs. WT, *p* < 0.001), and changed the morphology of the metastatic grade IV PC-3 cells from spindle fibroblast-like to rounded or polygonal (Figure 5C), as well as attenuated motility–migration (69.1% reduction vs. WT, *p* < 0.01) and invasion (83.3% reduction vs. WT, *p* < 0.001) after 24 h (Figure 5D). In addition, we observed that SHBG_OE or SRD5A1 knockout (sgSRD5A1) significantly suppressed the ability of metastatic PC-3 and LNCaP cells to form primary or primary-derived secondary tumorspheres (Figure 5E), with concomitant downregulation of HSD17B2, HSD17B3, stemness KLF4, OCT3/4, and drug resistance ABCC1, ABCB1 protein expression levels (Figure 5F). This expression profile was corroborated by results from bioinformatics-aided exploration and analysis of pharmacogenomic data related to cancer cell lines, including prostate cancer cell lines, across different sources from the NIH Genomics and Pharmacology Facility, Developmental Therapeutics Branch (DTB) (https://discover.nci.nih.gov/, accessed on 25 February 2021) which showed that, converse to SHBG, SRD5A1 is co-expressed with drug resistance ABCG2, ALDH1A1, ABCC2, and stemness OCT3/4, KLF4 markers (Figure 5G), and associated with insensitivity to Docetaxel activity in DU-145 and PC-3 PCa cell lines, using the NCI-60 Human Tumor Cell Lines Screen platform (https://dtp.cancer.gov/discovery_development/nci-60/, accessed on 24 February 2021) (Figure 5H). However, when SRD5A1 is targeted by CRISPR, criSRD5A1 elicited corresponding upregulation of SHBG with concurrent suppression of KLF4, POU5F1/OCT3/4, ABCC2, and ALDH1A1 in hormone refractory metastatic VCaP cells (Figure 5I). These data, in the context of genomics of drug sensitivity in cancer, provide some proof of concept that re-reprogramming of testosterone metabolism via “SRD5A1 withdrawal” or “SHBG induction” is a workable therapeutic strategy for shutting down androgen-driven oncogenic signals, reversing cancer stem cell (CSC)-facilitated treatment resistance, and repressing the metastatic/recurrent phenotypes of patients with PCa.

### 3.6. Compared with PSA, the HSD17B2/HSD17B3/SHBG/SRD5A1 4-Gene Signature Is Capable of Differentiating Recurrent/Nonresponsive from Nonrecurrent/Responsive PCa

Against the background of the diagnostic and prognostic relevance of PSA in urology practices, our statistical analysis of the TCGA PRAD cohort (n = 497) showed that, compared with selected clinicopathological variables, including PSA, the testosterone tetrad exhibited the largest effect size for prediction of therapy response or disease recurrence (Cohen’s *d*: HSD17B2 = 3.9, HSD17B3 = 7.0, SHBG = 11.9, SRD5A1 = 27.5 vs. PSA = 0.11) (Appendix A). In concordance, factor inclusion probability analysis using the Bayesian linear regression model showed that, while SRD5A1 (ENSG00000145545.10), SHBG (ENSG00000129214.13), HSD17B3 (ENSG00000130948.8), and HSD17B2 (ENSG00000086696.9) transcended the threshold for inclusion as predictors of recurrence or “new tumor event after initial treatment”, PSA value and age at diagnosis did not (Figure 6A). In parallel analyses, using K-means clustering, patients from the TCGA PRAD cohort (n = 497) were ably stratified into cluster 1 (nonrecurrent, therapy responsive) and cluster 2 (recurrent, therapy nonresponsive) by the testosterone tetrad and age at diagnosis, while PSA value exhibited no apparent differentiating potential (Figure 6B,C). Patient stratification based on t-distributed stochastic neighbor embedding (t-SNE)-aided dimensionality reduction of the TCGA PRAD dataset showed that combining the testosterone tetrad and age at diagnosis fairly differentiates the nonrecurrent, therapy responsive cases from the recurrent, therapy nonresponsive cases; however, it leaves ~22% of the cohort pooling away from their supposedly designated cluster (Figure 6D). Our profiling of these prediction variables using the generalized linear model (*X*^2^ = 14.71, *p* = 0.012) showed that age at diagnosis exhibited the smallest effect, and may be excluded (Prob > *X*^2^ = 0.535) (Figure 6E, *upper;* also see Appendix A). More so, receiver operating curve (ROC) analyses showed that the testosterone tetrad very well predicts “new tumor event after initial treatment” (Training: n = 407; AUC_CR_ 0.671; AUC_PR_ 0.696; AUC_PD_ 0.767; AUC_SD_ 0.665; Generalized R^2^ = 0.156; -Loglikelihood = 160.88 vs. Validation: n = 44; AUC_CR_ 0.698; AUC_PR_ 0.878; AUC_PD_ 0.950; AUC_SD_ 0.566; Generalized R^2^ = 0.305; -Loglikelihood = 16.32), and “follow-up treatment success” (Training: n = 385; AUC 0.722; Generalized R^2^ = 0.126; -Loglikelihood = 233.35 vs. Validation: n = 42; AUC 0.844; Generalized R^2^ = 0.178; -Loglikelihood = 22.81) in the TCGA PRAD cohort. Using the artificial neural network (ANN) model, an unsupervised machine learning algorithm, we showed that the differential expression of HSD17B2, HSD17B3, SHBG, and SRD5A1 may concertedly predict “new tumor event after initial treatment” and “follow-up treatment success” (Figure 6F), especially as the putative predictive potential of the testosterone tetrad was validated using the GSE40272 PCa dataset (n = 153), with the HSD17B2^lo^HSD17B3^hi^SHBG^lo^SRD5A1^hi^ group exhibiting significantly worse recurrence-free survival compared with their HSD17B2^hi^HSD17B3^lo^SHBG^hi^SRD5A1^lo^ counterparts (concordance index = 77.9%, log-rank equal curves *p* = 0.0008, R^2^ = 0.31/0.86, risk groups HR = 6.96 (95% CI: 1.9–25.47), *p* = 0.003) (Figure 6G).

### 3.7. Inhibition of SRD5A1 with Dutasteride Synergistically Enhances the Anticancer Potential of Low Dose Docetaxel, Reduced Metastatic Burden, and Confer Survival Advantage, In Vivo

To provide more insight into the mechanistic underlining and pathobiological implication of the differential but concerted alteration of the HSD17B2, HSD17B3, SHBG, and SRD5A1 testosterone tetrad in disease course and therapy response of patients with PCa, we further probed the discovery cohort of the GSE70768 dataset (n = 199), consisting of complete, quality-controlled HT12v4 data for 13 castration-resistant prostate cancer (CRPC), 113 tumor, and 73 matched benign samples. Consistent with earlier results, we observed that, compared with *HSD17B2* (t = 5.17, *p* = 0.00025) and *SHBG* (t = 0.54, *p* = 0.75) gene expression levels which were inversely correlated with the Gleason score, we found that *SRD5A1* was positively correlated with increased Gleason score, and this was statistically significant (t = 2.61, *p* = 0.028), while the association between *HSD17B3* expression and Gleason score was equivocal (Figure 7A). Using same PCa cohort data, we found that, while *SHBG* expression was downregulated in patients with CRPC compared with their peers in the tumor or benign group, howbeit statistically insignificant (t = 0.40, *p* = 0.67), relative to the benign group, *SRD5A1* expression was significantly upregulated in the tumor and CRPC groups (t = 11.39, *p* = 0.000021) (Figure 7B). Because of this demonstrated implication of the testosterone tetrad in disease aggressiveness and poor prognosis, performing an in silico protein-compound association query using the DrugBank database (https://go.drugbank.com/, accessed on 19 February 2021), we identified small molecule inhibitors of components of the testosterone tetrad, including the US FDA-approved SRD5A1 inhibitors, Dutasteride, and Finasteride (Figure 7C). Furthermore, we exposed androgen-independent metastatic prostate carcinoma PC-3 cells to 1–5 μM Docetaxel (DOC; US FDA-approved for hormone-refractory metastatic PCa) with or without 1–5 μM Dutasteride (DUT). Drug-response/effect analyses show that combining DOC with DUT significantly enhanced the pharmacologic activity of DOC (~3.3-fold, *p* < 0.001), and that this effect was largely synergistic in nature using the CompuSyn-generated isobologram and confirmed by a zero interaction potency (ZIP) synergy score of 13.24, where a score > 10 implies synergism, −10 to 10 represents additivity, and <−10 means antagonism, using the SynergyFinder web application version 2.0 (https://synergyfinder.fimm.fi/, accessed on 26 January 2021) (Figure 7D). Evaluating the replicability of these findings in vivo (Figure 7E), we demonstrated that, compared with the control mice bearing PC-3 WT cells, significant reduction in the tumor burden of the PC-3_SHBG_OE (1.81-fold, *p* < 0.001) or PC-3_sgSRD5A1 (1.87-fold, *p* < 0.001) mice was observed by day 43. Pharmacological-wise, by day 43, treatment with DUTDOC combo had elicited a 2.28-fold reduction in tumor burden (*p* < 0.0005) (Figure 7F). In addition, we demonstrated that, in comparison to the control mice, the PC-3_SHBG_OE and PC-3_sgSRD5A1 bearing mice exhibited a 40 or 60% survival advantage, respectively, while treatment with the DUTDOC combo conferred an 80% survival advantage (*X*^2^ = 60.20, *p* < 0.0001) (Figure 7G). Similarly, we found that, compared to the control mice, the PC-3_SHBG_OE, PC-3_sgSRD5A1, and DUTDOC combo mice enjoyed a 50, 80, or 90% metastasis-free survival advantage (*X*^2^ = 60.73, *p* < 0.0001) (Figure 7H), and this was consistent with the number of metastatic nodules found on excised lungs, livers, brains, and bones of the tumor-bearing mice (Figure 7I). These results show that the testosterone tetrad plays an important role in the tumor growth, disease progression, and therapy response of highly metastatic PCa cells in vivo, and indicate, at least in part, that the pharmacological targeting of SHBG or SRD5A1 synergistically enhance the therapeutic effect of conventional anticancer therapy, in vivo.

## 4. Discussion

In the last three decades, our increased understanding of tumor biology has increasingly aided the unraveling of appreciable metabolic complexity of malignant cells and cancerization, regardless of cancer type or histological sub-type [17,18]. It is of translational relevance that certain cancerous cells are more dependent on specific oncogenic, metabolic, or oncometabolic pathways for survival than others within this evolving metabolic landscape. For instance, it is well established that the activation of AR is essential for the growth of prostate cancer; more so, castration-resistant prostate cancer (CRPC) is largely addicted to functional AR, with several mechanisms suggested to explain this addiction, including AR point mutations, gene amplification, and/or overexpression [19]. Interestingly, there is accruing evidence of the enhanced conversion of the weak adrenal androgen called androstenedione to testosterone by steroidogenic effectors of androgen metabolism in patients with metastatic CRPC compared to primary treatment-naïve PCa [20]. Upregulated androgenic enzymes within the tumor bulk facilitates enhanced conversion of adrenal steroids into gonadal steroids like testosterone and DHT, with associated AR-mediated transcriptional activation consequently driving castration resistance [19,20]. The present study demonstrated that the differential expression of HSD17B2, HSD17B3, SHBG, and SRD5A1 in patients with different PCa grades is suggestive of a dysregulated androgenic-signal-driven oncogenicity in patients with PCa, where intracellular signaling skewed towards a dominant SRD5A1/HSD17B3 at the expense of SHBG/HSD17B2 signaling drives enhanced cancerization, and may inform therapeutic decision making and management strategy for patients with PCa (Figure 1). This is therapeutically relevant within the context of the centrality of ADT to the management of metastatic and recurrent PCa, as well as the critical role of androgen hormones and androgenic signaling in the development and progression of PCa, especially as CRPC, once attributed to hormone refraction, is increasingly shown to be driven by sex steroid hormones [21,22]. In partial congruence with our findings, Khvostova et al. reported that, compared to normal adjacent prostate tissues, the expression of HSD17B3 and SRD5A2 was markedly increased in benign prostatic hyperplasia (BPH) tissues, HSD17B2 expression was significantly decreased in all samples, and SRD5A2 transcript level was upregulated in BPH and PCa compared to the normal adjacent prostate tissues [23]. More so, our finding is consistent with contemporary knowledge that human HSD17B2 suppresses the precursors of testosterone and DHT, namely C_19_17-ketosteroids dehydroepiandrosterone (DHEA), 5α-androstanedione, and androsterone, while HSD17B3-mediated conversion of DHEA to androst-5-ene3β,17β-diol contributes to upregulated synthesis of testicular testosterone [23,24].

Furthermore, this study provides some pre-clinical evidence that the nonrandom, but concerted and well-calibrated interplay between the HSD17B2/HSD17B3/SHBG/SRD5A1 tetrad determines therapy response, facilitates disease progression and is relevant in clinical prognosis, especially with disease-driving roles attributed to SHBG and SRD5A1, while HSD17B2 and HSD17B3 are complicit passengers (Figure 2 and Figure 3). This is consistent with reports of inversely correlated serum testosterone and SHBG levels, with proposed association between high serum SHBG and moderate decrease in the risk of PCa [25], as well as suggestions in a recent study that serum testosterone levels during ADT may be prognostic of the progression to CRPC in patients with metastatic PCa [26]. More so, “recent assessment of CRPC cells has identified increased expression of steroidogenic enzymes such HSD3B1, HSD3B2, HSD17B3, AKR1C3, and SRD5A1, which may contribute to *de novo* production of steroids and androgens” [27].

For the first time, to the best of our knowledge, we also provided some evidence of a therapeutically actionable direct interaction between the testosterone tetrad, and that this distinct interaction is not just spatiotemporal, but functionally primed to drive cell cycle progression, enhance motility, cancer stemness, and resistance to therapy (Figure 4). In concordance with our findings that a high SRD5A1/SHBG ratio is associated with upregulated drug efflux proteins ALDH1A1, ABCB1, ABCC1, ABCG2, and markers of pluripotency, including KLF4, LIN28, OCT4, and SOX2, there are reports implicating enhanced drug-efflux in the resistance of CRPC to many chemotherapeutics, including Docetaxel, a drug of choice for hormone-naïve advanced PCa in conjunction with ADT [27]. In fact, upregulated ABCB1 has been associated with the acquisition of a CRPC phenotype regardless of androgen-dependence status, as demonstrated by overexpression of ABCB1 in both the moderately metastatic, PSA negative, and hormone-insensitive DU145 and PSA positive, hormone-sensitive 22RV1 cell lines when made Docetaxel-resistant [27,28]. Other mechanisms of therapy resistance include dysregulated activity of cell cycle regulators, often overexpressed in PCa. In line with this, our study also found high SRD5A1/SHBG ratio to be strongly associated with upregulated expression of CDK1 (the only CDK that is essential for cell cycle progression) and AURKA, a dyad that is characteristic of uncontrolled proliferation of cancerous cells, especially as the activation of CDK1 in complex with cyclin A/B is required and sufficient for entry into mitosis [29]. AURKA, which is localized in the centrosome during the S phase, is a serine/threonine kinase that plays a critical role in mitosis and cytokinesis, particularly because of its indispensability in centrosome maturation, spindle assembly, and spindle orientation [29]. Furthermore, in addition to promoting the activation of CDK1 and mitotic entry, especially after G2 phase arrest due to DNA damage, AURKA also prevents the proteasomal degradation of N-MYC, thus stabilizing it, and consequently promoting G1-S progression [29,30]. The association of increased SRD5A1/SHBG ratio with upregulated expression of stemness markers LIN28, NANOG, KLF4, OCT4, and SOX2 is therapeutically relevant in terms of cancer stem-cell-targeting strategy, as it suggests an alternative approach to inhibiting signaling that are essential for the homeostasis and function of cancer stem cells, which are well documented drivers of cancer progression. This is particularly important because these stemness markers are implicated in dysregulated cell cycle, aberrant cell proliferation, enhanced cancer aggression, evasion of death signals, multidrug resistance, and worse prognosis [31].

Furthermore, contextualized within the genomics of cancer drug sensitivity, we also provided some proof of concept that re-reprogramming of testosterone metabolism via “SRD5A1 withdrawal” or “SHBG induction” is a workable therapeutic strategy for shutting down androgen-driven oncogenic signals, reversing cancer stem cell (CSC)-facilitated treatment resistance, and repressing the metastatic/recurrent phenotypes of patients with PCa (Figure 5). This is clinically significant as it further validates the burgeoning anticancer therapeutic concept that oncogene-addicted malignancies, including testosterone-driven PCa, present valuable actionable targets for therapeutic intervention with enhanced chance of achieving wide therapeutic windows. Our study demonstrates that aggressive PCa cells, especially metastatic and/or recurrent CRPC cells, exhibit a preferential dependence on androgen-driven oncogenic signaling pathway for maintaining their malignant phenotype, survival, and proliferation, regardless of the characteristic broad array of impairments within their genetic make-up, thus revealing a promising Achilles’ heel of PCa [32]. We posit that the withdrawal of SRD5A1 or induction of SHBG in aggressive PCa cells elicits “oncogenic shock” [33], which entails the death of the testosterone tetrad-addicted cancerous cells upon inhibition of the oncoprotein SRD5A1 to which they are seemingly addicted, or ectopic expression of the tumor suppressor SHBG to which they were estranged. This position is supported by our data showing significantly suppressed PC-3 cell viability, proliferation, motility (migration and invasion), and primary and secondary tumorsphere formation, with associated downregulation of TWIST1, SLUG, SNAIL, VIM, KLF4, OCT3/4, ABCB1, and ABCC1 upon “SRD5A1 withdrawal” by CRISPR knockout of SRD5A1, or “SHBG induction” using SHBG overexpression. The observed downregulation of both stemness and EMT markers by targeting components of the testosterone tetrad is consistent with contemporary knowledge of the complex interplay between cancer stemness and EMT. In concordant reports, it was recently demonstrated that α2β1integrin^hi^ CD133+ prostate basal stem cells are characterized by enhanced expression of the transcription factor Zeb1, a vital inducer of EMT, both in silica, in vitro, and in vivo [34,35]. Corollary to this, it has also been reported that these “multipotent basal stem cells are exclusively located in juxta-urethral niches and function in a directed migratory way to generate epithelial progenitors” [34], which is reminiscent of CSC-associated self-renewal, repopulation of cancerous prostate cells, and disease relapse after initial response. More so, PCa cells with mesenchymal traits have been shown to exhibit stem-cell-like phenotypes, such as upregulated expression of pluripotency factors OCT2, SOX2, and NANOG, with concomitant increase in clonogenicity, tumorsphere formation capability, and tumorigenicity in vivo [36]. Together, these accentuate the translational relevance of our results indicating that the molecular components of the testosterone tetrad are actionable, and may be exploited as an efficacious therapeutic strategy in patients with metastatic or recurrent PCa.

It is also therapeutically relevant that, compared with PSA, the HSD17B2/HSD17B3/SHBG/SRD5A1 4-gene signature is capable of differentiating recurrent/nonresponsive from nonrecurrent/responsive PCa (Figure 6). Our findings further highlight the significant shortcomings of PSA-based decision-making in PCa, despite the fact that PSA measurement remains common practice at various stages of PCa management, including screening or assessing potential risk of developing PCa, detection of recurrence after initial therapy, and for informing the management of advanced disease [37]. Interestingly, accruing evidence indicate that “PSA screening does lead to over-diagnosis, over-treatment, and treatment-associated morbidity” [38], and, consistent with the European Association of Urology (EAU) PCa guidelines panel recommendations, the PSA-based “BCR after primary treatment of localized prostate cancer does not necessarily lead to clinically apparent progressive disease” [39]. This study proffers an alternative multi-gene tool (HSD17B2^lo^HSD17B3^hi^SHBG^lo^SRD5A1^hi^ vs. PSA^hi^) for identifying patients at high risk of progression or relapse to initiate early salvage treatment, while deferring treatment for patients with low risk of recurrence or progression.

Moreover, exploiting the phenomenon of oncogene addiction, we demonstrated for the first time, to the best of our knowledge, that the inhibition of SRD5A1 with metronomic Dutasteride synergistically enhances the anticancer potential of low dose Docetaxel, reduced metastatic burden, and confers survival advantage, in vivo (Figure 7). These findings bear therapeutic relevance considering that the majority of patients with PCa develop resistance to therapy, including Docetaxel, and progress, despite initially responding well to treatment [40]. Resistance or reduced sensitivity to Docetaxel is a complex biological phenomenon that largely depends on the activation of several intratumoral and paratumoral molecular signaling pathways, including the androgenic/steroidogenic signals. Dutasteride-induced pharmacological withdrawal of SRD5A1, with its associated mechanistic underpinnings, effectively elicits an impressive preclinical therapeutic activity consistent with those recently observed in clinical practice “following treatment with so-called ‘rationally-targeted’ agents” [32,33]. The preclinical response was remarkable, and reminiscent of clinical responses to such single agents in patients with metastatic PCa that were largely refractory to standard chemotherapy [41,42].

## 5. Conclusions

Over the last decade or two, our improved understanding of the pathobiology of androgen-addicted PCa and documented therapeutic advances/breakthroughs have not translated into any substantial or sustained curative benefit for patients treated with traditional ADT or novel immune checkpoint blockade therapeutics. This is invariably connected with the peculiar biology and intratumoral heterogeneity of PCa. CRPC, constituting ~30% of all PCa, remains a clinically enigmatic and therapeutically challenging disease sub-type that is therapy-refractory and characterized by high risk for recurrence after initial response.

Our findings highlight the role and exploitability of testosterone metabolic reprogramming in prostate TME for patient stratification and personalized/precision medicine based on the differential but concerted expression of molecular components of the proposed testosterone tetrad in patients with therapy-refractory, locally advanced, or recurrent PCa. The therapeutic exploitability and clinical feasibility of our proposed approach is suggested by our preclinical findings. However, validation using a large heterogeneous multicenter clinical cohort is warranted.

## Figures and Tables

**Figure 1 cancers-13-03478-f001:**
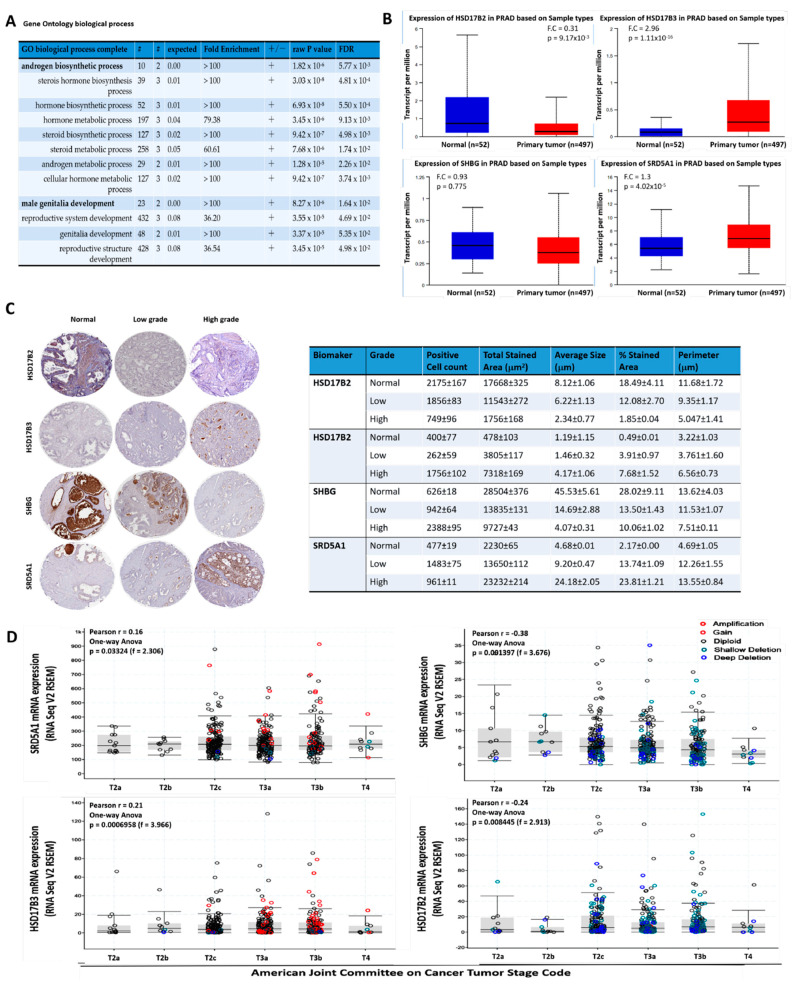
The aberrant expression of the testosterone metabolic tetrad HSD17B2, HSD17B3, SHBG, and SRD5A1 characterize androgen-addicted PCa and is associated with disease progression. (**A**) Depiction of the gene ontology biological-process-based Fisher’s exact overrepresentation test using the GO Ontology database. (**B**) Histograms showing the differential expression of HSD17B2, HSD17B3, SHBG, or SRD5A1 mRNA in normal and primary tumor from the TCGA PRAD cohort. (**C**) Representative photo-images and quantitative chart of the expression of HSD17B2, HSD17B3, SHBG, or SRD5A1 protein in normal prostate, low-grade, or high-grade samples from the SHH PCa cohort. (**D**) Box and whisker plots showing the differential expression of SRD5A1, SHBG, HSD17B2, or HSD17B3 mRNA according to the AJCC tumor stage. FDR, false discovery rate; F.C, fold change; SHH, Shuang Ho Hospital; AJCC, American Joint Committee on Cancer.

**Figure 2 cancers-13-03478-f002:**
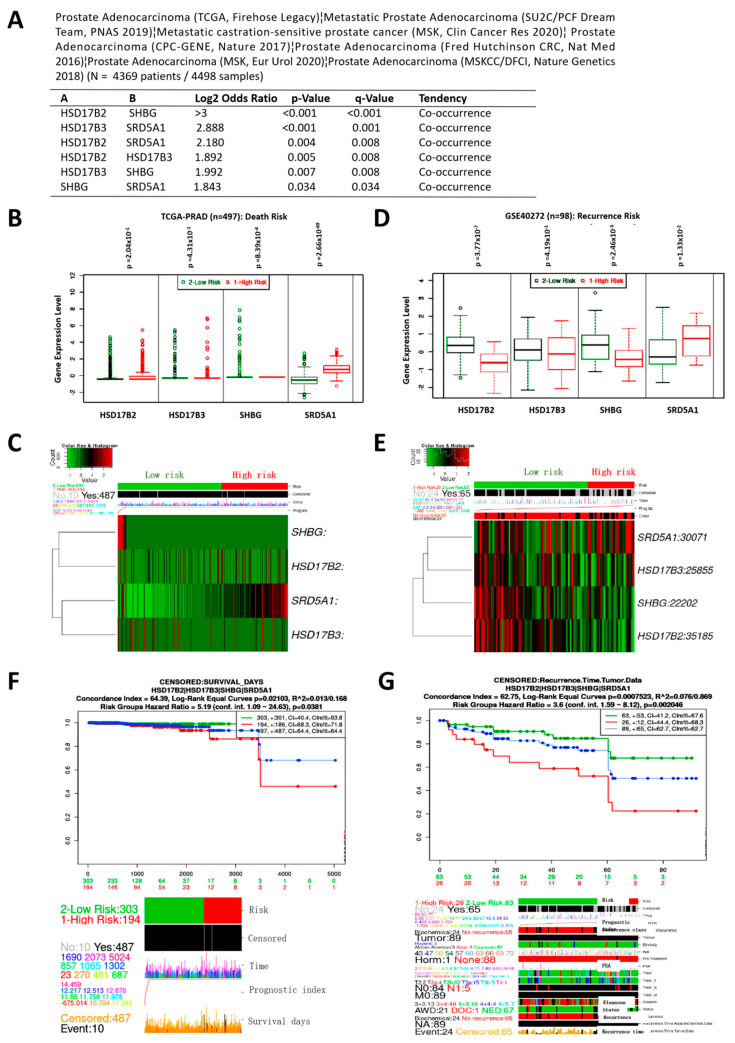
Variation in HSD17B2, HSD17B3, SHBG, and SRD5A1 expression co-occur and concertedly bear significant prognostic relevance in patients with PCa. (**A**) Depiction of mutual exclusivity or co-occurrence analysis using a multi-source pooled primary and metastatic PRAD cohort (N = 4369 patients/4498 samples). (**B**) Graphical representation and (**C**) heatmap of the association between HSD17B2, HSD17B3, SHBG, or SRD5A1 gene expression and death risk in the TCGA PRAD cohort. (**D**) Graphical representation and (**E**) heatmap of the association between HSD17B2, HSD17B3, SHBG, or SRD5A1 gene expression and recurrence risk in the GSE40272 cohort. Kaplan–Meier curves of the (**F**) overall survival and (**G**) recurrence-free survival based on median-dichotomized, concerted HSD17B2/HSD17B3/SHBG/SRD5A1 expression. PRAD, prostate adenocarcinoma.

**Figure 3 cancers-13-03478-f003:**
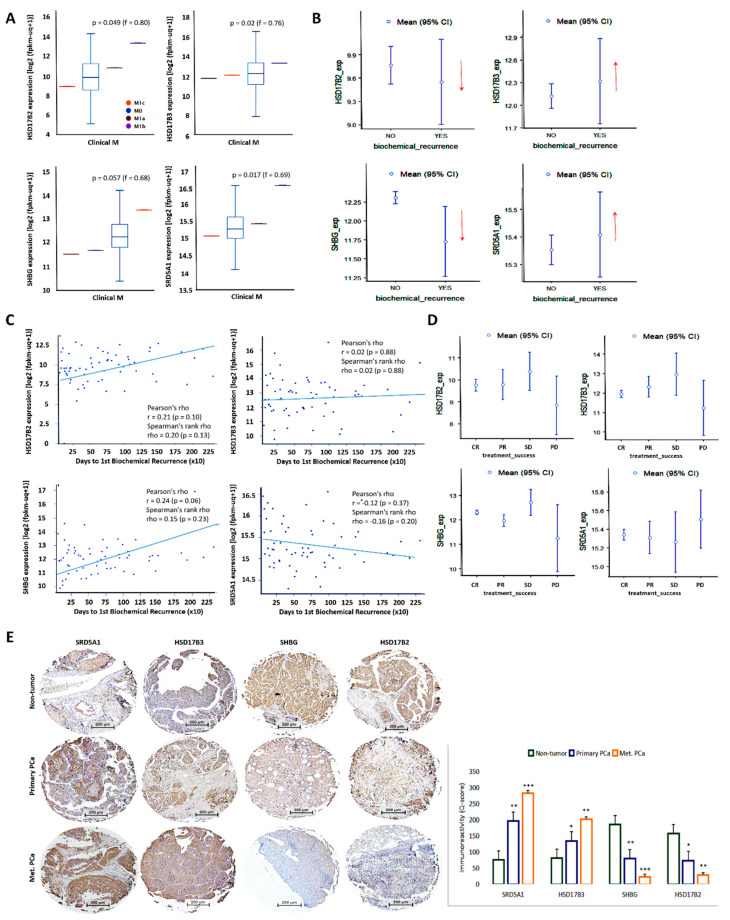
The differential but concerted expression of HSD17B2, HSD17B3, SHBG, and SRD5A1 is associated with the metastatic and recurrent phenotype of patients with PCa. Graphical representation of the association between HSD17B2, HSD17B3, SHBG, or SRD5A1 mRNA expression level and (**A**) clinical M stage, (**B**) biochemical recurrence status, (**C**) number of days to biochemical recurrence, and (**D**) treatment success, in the GDC TCGA PRAD cohort. (**E**) Representative photo-images and histograms of SRD5A1, HSD17B3, SHBG, or HSD17B2 protein expression level in adjacent nontumor prostate, primary, or metastatic samples from the SHH PCa cohort. M, metastasis; exp, expression; CI, confidence interval; GDC, genomic data commons; SHH, Shuang Ho Hospital; *, *p* < 0.05; **, *p* < 0.01; ***, *p* < 0.001.

**Figure 4 cancers-13-03478-f004:**
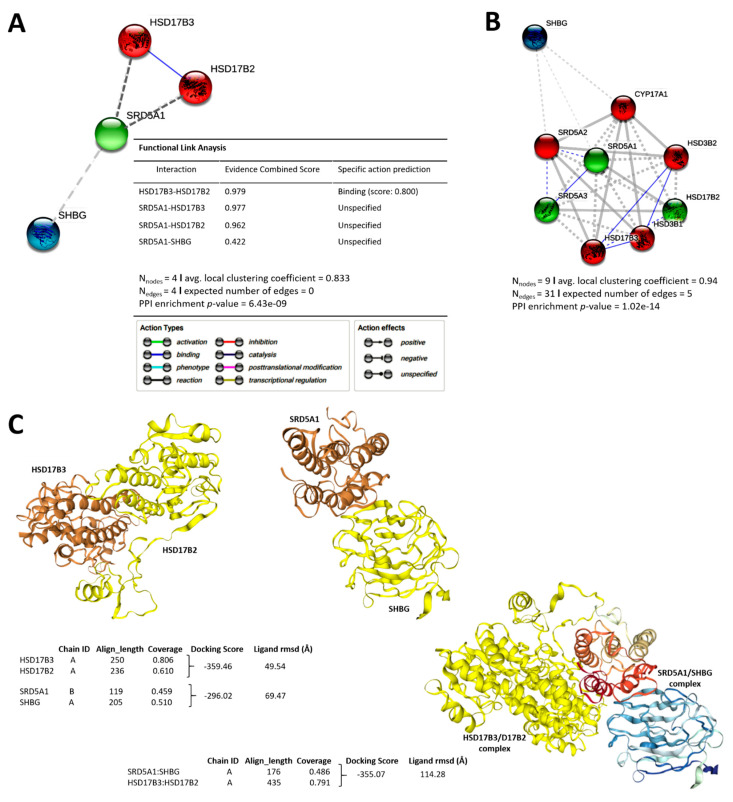
Distinct interaction between the testosterone tetrad elicits androgenic signals that drive cell cycle progression, enhanced motility, cancer stemness, and resistance to therapy in patients with PCa. STRINGdb-generated visualization of the protein–protein interaction between (**A**) HSD17B2, HSD17B3, SHBG, and SRD5A1, as well as with (**B**) SRD5A2, SRD5A3, HSD3B1, HSD3B2, and CYP17A1. (**C**) Molecular docking showing the direct interaction between HSD17B2 and HSD17B3, SRD5A1 and SHBG (upper left), as well as protein tetrad formation by the HSD17B2/HSD17B3 and SRD5A1/SHBG complexes (lower right). Heatmaps showing the correlation between expression profile of the testosterone tetrad and molecular effectors of (**D**) multidrug resistance, and (**E**) cancer stemness, in the GSE26910 mixed prostate and breast cancer cohort. Columns with similar annotations are collapsed by taking mean inside each group. Rows are centered; unit variance scaling is applied to rows. Both rows and columns are clustered using correlation distance and average linkage. (**F**) Heatmap showing the correlation between expression profile of the testosterone tetrad and ABCC1, ABCB1, ABCG2, ALDH1A1 expression in GSE33455 docetaxel-resistant PCa cell lines dataset. (**G**) Transcriptional profiling heatmap showing the correlation between the differential expression of the testosterone tetrad in human benign prostatic hyperplasia, primary and metastatic PCa samples from the GSE3325 PCa cohort. (**H**) Molecular docking showing the interaction cascade between the testosterone tetrad and molecular effectors or mediators of cell cycle progression, and drug uptake/therapy response.

**Figure 5 cancers-13-03478-f005:**
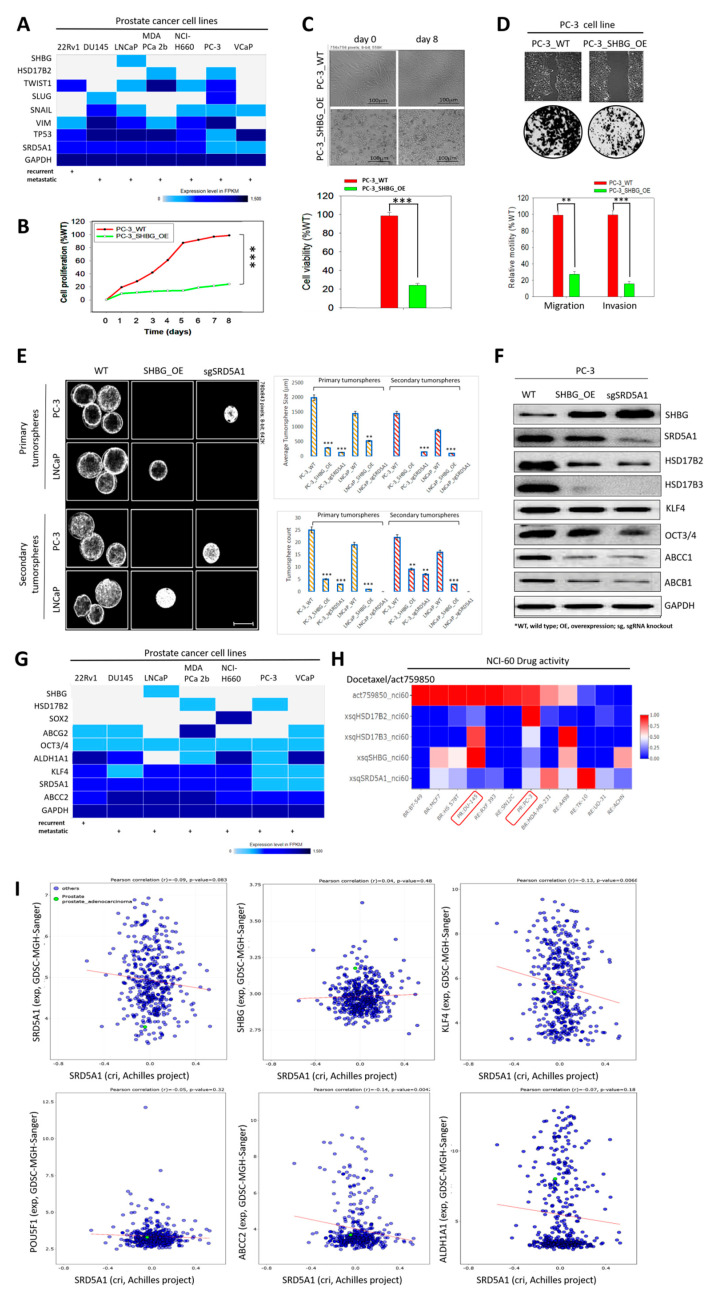
Molecular fine-tuning of components of the 4-gene PCa signature modulate the highly proliferative, metastatic and cancer stem-cell-like (cum disease recurrent) phenotypes of PCa cells. (**A**) Heatmap of the differential expression of SHBG, HSD17B2, TWIST1, SLUG, SNAIL, VIM, TP53, SRD5A1, and GAPDH in the 22Rv1, DU145, LNCaP, MDA-PCa 2b, NCI-H660, PC-3, and VCaP prostate cancer cell lines from the NIH GDC Cancer Cell Line Encyclopedia. (**B**) Line graph showing the effect of SHBG_OE on PC-3 cell proliferation over 8 days. (**C**) Photo-images (upper) and histograms (lower) of the effect of SHBG_OE on PC-3 cell viability on day 0 and day 8. (**D**) Photo-images (upper) and histograms (lower) of the effect of SHBG_OE on PC-3 cell migration or invasion. (**E**) Photo-images and histograms showing the differential effects of SHBG_OE or sgSRD5A1 on the formation of primary or secondary tumorspheres by PC-3 or LNCaP cells. (**F**) Representative Western blot images of the differential effects of SHBG_OE or sgSRD5A1 on the expression levels of SHBG, SRD5A1, HSD17B2, HSD17B3, KLF4, OCT3/4, ABCC1, and ABCB1 in PC-3 cells. Original figure see Appendix A. (**G**) Heatmap of the differential expression of SHBG, HSD17B2, SOX2, ABCG2, OCT3/4, ALDH1A1, KLF4, SRD5A1, ABCC2, and GAPDH in the 22Rv1, DU145, LNCaP, MDA-PCa 2b, NCI-H660, PC-3, and VCaP prostate cancer cell lines from the NIH GDC Cancer Cell Line Encyclopedia. (**H**) Heatmap of the differential pharmacological effect of Docetaxel on HSD17B2, HSD17B3, SHBG, or SRD5A1 expression in the NCI-60 Human Tumor Cell Lines. Red box shows PCa cell lines. (**I**) Dot and line plots showing the correlation between SRD5A1 knockout by CRISPR and SRD5A1, SHBG, KLF4, POU5F1/OCT4, ABCC2, or ALDH1A1 expression in the GDSC–MGH–Sanger cohort. GAPDH is loading control. GDSC, the genomics of drug sensitivity in cancer project; MGH, Massachusetts General Hospital; Sanger—the Wellcome Sanger Institute (UK). %, percentage; WT, wild type; OE, overexpression; **, *p* < 0.01; ***, *p* < 0.001.

**Figure 6 cancers-13-03478-f006:**
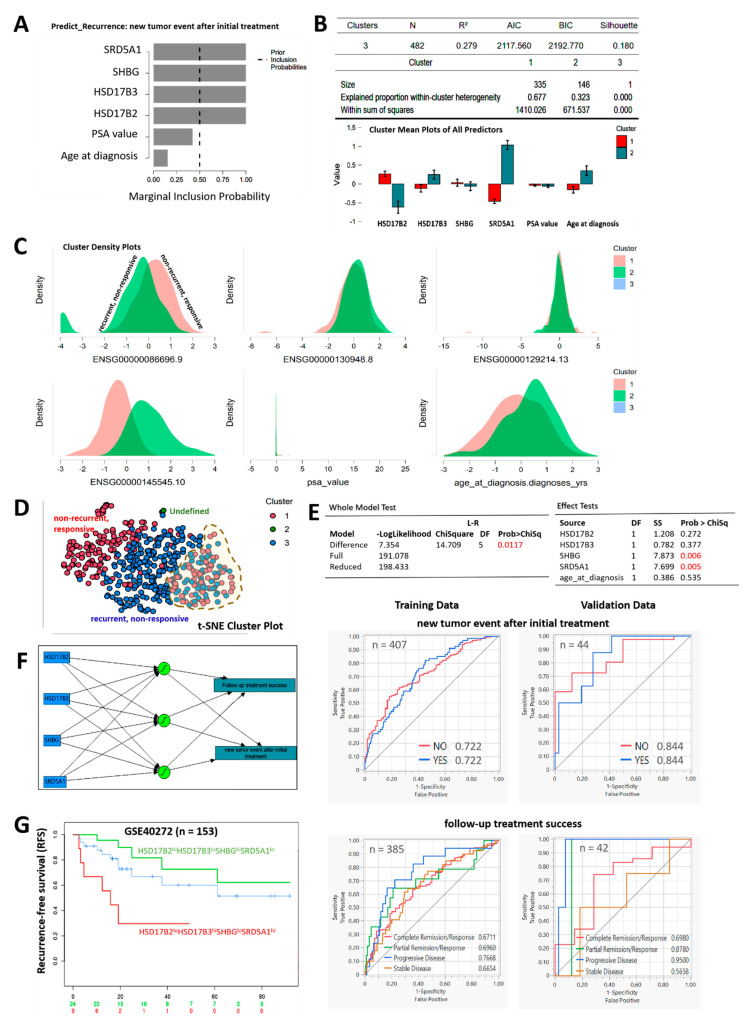
Compared with PSA, the HSD17B2/HSD17B3/SHBG/SRD5A1 4-gene signature is capable of differentiating recurrent/nonresponsive from nonrecurrent/responsive PCa. (**A**) Graphical representation of the probability of including SRD5A1, SHBG, HSD17B2, HSD17B3, PSA value, or age at diagnosis as predictors of recurrence/new tumor event after initial treatment. (**B**) Histograms of cluster mean plots of all potential predictors. (**C**) Cluster density plots using HSD17B2 (ENSG00000086696.9), HSD17B3 (ENSG00000130948.8), SHBG (ENSG00000129214.13), SRD5A1 (ENSG00000145545.10), PSA value, and age at diagnosis. Clustering is based on k-means. (**D**) t-SNE cluster plot showing stratification of the TCGA PRAD cohort into nonrecurrent, responsive or recurrent (cluster 1), nonresponsive clusters (cluster 2), and undefined (cluster 3). (**E**) Prediction profiler based on generalized linear model showing correlation between HSD17B2, HSD17B3, SHBG, SRD5A1, or age at diagnosis and new tumor event after initial treatment (upper). AUROC analysis showed that the testosterone tetrad predicts new tumor event after initial treatment (middle), and follow-up treatment success (low). AUROC, area under the receiver operating curve; t-SNE, t-distributed stochastic neighbor embedding. (**F**) Pictorial visualization of the Artificial neural network (ANN) model, showing the input nodes HSD17B2, HSD17B3, SHBG, and SRD5A1 interacting to generate the output nodes “new tumor event after initial treatment” and “follow-up treatment success”. Three hidden nodes are empirically generated. (**G**) Kaplan-Meier curve showing the differential effect of HSD17B2^lo^HSD17B3^hi^SHBG^lo^SRD5A1^hi^ and HSD17B2^hi^HSD17B3^lo^SHBG^hi^SRD5A1^lo^ on the recurrence-free survival in the TCGA PRAD cohort.

**Figure 7 cancers-13-03478-f007:**
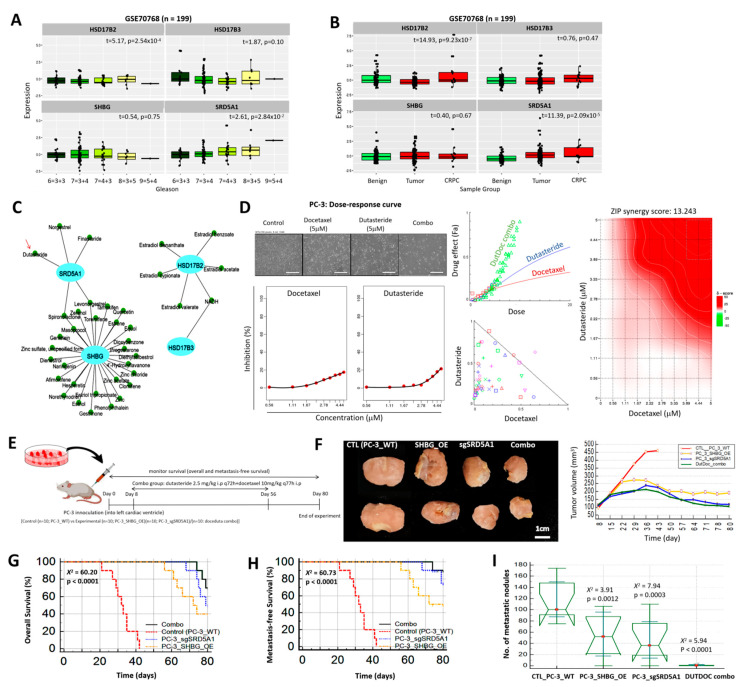
Inhibition of SRD5A1 with Dutasteride synergistically enhances the anticancer potential of low dose Docetaxel, reduces metastatic burden, and confers survival advantage, in vivo. Graphical representation of the correlation between HSD17B2, HSD17B3, SHBG, or SRD5A1 and (**A**) Gleason score, or (**B**) sample group in the GSE70768 PCa cohort. (**C**) Visualization of the in silico protein-compound association query using the DrugBank database. (**D**) Photo-images and dose–response curves showing individual and synergistic effects of Docetaxel and/or Dutasteride on the proliferation/viability of PC-3 cells. (**E**) Schema of our tumor xenograft in vivo model. (**F**) Photo-image and line graph of the effect of SHBG_OE, sgSRD5A1, or docetaxel–dutasteride combination treatment on tumor volume in mice inoculated with PC-3 cells. Kaplan–Meier curves showing the effect of SHBG_OE, sgSRD5A1, or docetaxel–dutasteride combination treatment on the (**G**) overall, and (**H**) metastasis-free survival of mice inoculated with PC-3 cells. (**I**) Notched box and whiskers plot of the effect of SHBG_OE, sgSRD5A1, or docetaxel–dutasteride combination treatment on the number of metastatic nodules in mice inoculated with PC-3 cells.

## Data Availability

The datasets used and analyzed in the current study are publicly accessible as indicated in the manuscript.

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
