# Peer review of "Differential but Concerted Expression of HSD17B2, HSD17B3, SHBG and SRD5A1 Testosterone Tetrad Modulate Therapy Response and Susceptibility to Disease Relapse in Patients with Prostate Cancer"

_cancers, 2021, doi:10.3390/cancers13143478_

Round 1
Reviewer 1 Report
In this study the authors have explored the feasibility of utilizing ‘testosterone tetrad’ as a surrogate biomarker in PCa diagnosis, metastasis, and recurrence. Using large cohorts, they demonstrate a clear correlation of these gene and protein expression with age, new tumor event, metastasis status, and Gleason score in patients, and further functionally demonstrate the effect on stemness and migration/invasion properties in cells, and tumor-formation, metastasis, survival, and drug synergy in mice xenograft models. The introduction and discussion are beautifully written, the experiments are aptly designed, executed, and explained. The observations of this study will be highly beneficial to the research and clinical community.
The manuscript can be accepted after addressing few minor queries.
- The spelling/grammar needs to be checked in few places. Some examples below:
Line 96: increased
166: prediction of disease course
177: have been characterized
178: depends on
Lines 200-207: very long sentence
382: placed
- Line 107 mentions ‘Pictorial Abstract’ which I could not find.
- Methods says all cells were used at p<2, does this apply to cell lines like DU145 and PC3 too or just primary cells?
- Text in some figures can be enlarged as currently they are not legible (like Figure 1B, 1D etc).
- As the manuscript contains numerous graphs of different kinds, in the legend it might be helpful for viewers if there is short description of how to read/interpret the graph (at least for ‘not so common’ graphs).
- It is quite well-established that the stemness properties correlate with epithelial-mesenchymal transition in various cancers including PCa. I am curious if authors would like to include some discussion on that?
- In Figure 5, what does %WT in Y-axis mean? I was bit confused as there is a line/bar for WT too.
- Please provide higher magnification images for Figure 5C.
Reviewer 2 Report
In this manuscript, the authors query if the genes HSD17B2, HSD17B3,SHBG and SRD5A1 play a role in prognosis and therapy resistance in prostate cancer patients. 1. The introduction is not concise or clear. The authors should adopt a funnel approach starting from the big picture and then moving step wise to the why these genes need to be investigated. I am still not clear why the authors chose to investigate these four particular genes. 2. Figure 1C, how many patient samples were stained for IHC? The quantification graphs are missing. HSD17B2 IHC changes are not apparent. 3. Figure 1D, While the amplification and deletion changes are evident with the red and blue circles, I am not sure if there is any correlation with T stage and levels of thee four genes. 4. The authors refer to a tetrad, yet they look at these genes one at a time. Did the authors look at how HSD17B2low SHBGlowHSD17B3highSRD5A1high as a group lplays a role in prognosis or therapy resistance. 5. Figure 3A, order of M1c, M0, M11 and M1d are inconsistent between genes. 6. The M1a and M1b groups have high expression of HSD17B2. It seems inconsistent with the story. Also SHBG is high in M1c. The authors do not explain what M1c classification means. 7. For the knock-down and overexpression studies, first, the authors do not show proof of knockdown or over-expression. Second, they only look at two of the four genes. Would be nice to see a comprehensive analysis for all four genes. 8. For 5C, the authors need to present a high magnification image to illustrate the change in morphology. 5E need to be quantified along with the replicates. 9. 6C, are the first four histogram for the four genes? 10. For the tumor studies, the authors are missing vehicle treatment and the mono treatment groups . How many n’s were used for 7F? it looks as if 2 mice per group was used. 11. 7B, the data does not reflect the text and those are moderate changes at best. Overall while this is an interesting hypothesis, the authors have to build a clear story and substantiate it with data.Author Response
Please kindly see the attachment
